# Effect of Nurses’ Preceptorship Experience in Educating New Graduate Nurses and Preceptor Training Courses on Clinical Teaching Behavior

**DOI:** 10.3390/ijerph18030975

**Published:** 2021-01-22

**Authors:** Kyung Jin Hong, Hyo-Jeong Yoon

**Affiliations:** 1Department of Nursing, Semyung University, Jecheon 27136, Korea; 2School of Nursing, Yeungnam University College, Daegu 42415, Korea; hjyoon@ync.ac.kr

**Keywords:** preceptor, preceptorship training course, satisfaction with training, clinical teaching behavior

## Abstract

Only few studies have examined the preceptor training courses and their effects on clinical teaching behaviors (CTBs) of preceptors. This study investigated preceptors’ experiences in educating new graduate nurses and the effect they had on clinical teaching behavior (CTB) based on whether they participated in a preceptor training program. A descriptive online survey method was used, and the participants included 180 registered nurses who were preceptors. The Clinical Teaching Behavior Inventory (CTBI-22) was used, and perceptions of teaching experiences were measured by six items. Data were analyzed using multiple linear regression. Preceptors working at general hospitals or hospitals were less likely to have participated in a preceptor training program than those working at tertiary hospitals. The overall mean score of CTB was 89.30, and “guiding inter-professional communication” showed the lowest mean score. Positive perceptions of preceptorship experiences were positively related with CTB, and the number of precepting experiences affected CTB only for nurses having undergone preceptor training courses. The use of role-playing as a method in training courses positively affected preceptors’ CTB. These findings suggest that preceptors need support from nurse managers and colleagues, and preceptor training programs should be developed.

## 1. Introduction

Nurses constitute the majority of medical personnel, and the high turnover and instability of nurse staffing are important study topics because they affect patient outcomes. In Korea, 42.7% of new graduate nurses who joined hospitals in 2017 were found to leave within a year [1]. Furthermore, systematic education and the support of co-workers and organization are imperative to maintain the current nurse workforce. 

Preceptorship is extensively used to help new graduate nurses adapt to hospitals and to educate them. Many health care facilities use the preceptorship system to facilitate new graduate nurses’ adjustment to their new roles and to bridge the gap between their learnings and real-world practices [2]. Newly graduated nurses’ turnover intention was negatively correlated with the preceptor’s transactional leadership, emotional leadership, preceptor–preceptee exchange relationship, and preceptee’s self-leadership [3]. Nevertheless, there are many cases where preceptors take on their role without receiving proper training on teaching new nurses [4]. The lack of training for preceptors could affect their ability to teach new graduate nurses and could have a negative effect on these nurses’ socialization in the clinical environment. Therefore, it is necessary to review the current status of preceptor training for teaching new graduate nurses and examine the impact of preceptors’ completion of training. 

A preceptor is “an educator who gives on-the-job training to novice nurses and nursing students”. As experienced nurses with specialized knowledge in their area of work, preceptors can aid novice nurses in adjusting to the clinical environment and impart to them specialized knowledge needed in their work environment [5]. The stress level of novice nurses, who interacted with their mentors on a regular basis, was found to decrease [6]. Research that examined the effect of training given by preceptors revealed that preceptor-based training programs increased the training satisfaction and retention rate of new nurses [7], and the caring of preceptors had a positive correlation with the job satisfaction and competence of novice nurses [8]. Therefore, providing training to newly graduated nurses, wherein preceptors maintain good relationships with new nurses and help them gain a positive perception regarding nursing work, is crucial to the socialization of new nurses. The roles of a preceptor include being an educator, evaluator, protector, and role model [9]. 

Nevertheless, many preceptors in health care facilities have difficulties due to their excessive workload, because they have to train new nurses and attend to patients at the same time. Previous studies have suggested that health organizations must train preceptors to effectively interact with newly graduated nurses to reduce turnover intention [10,11,12]. However, in some cases, they became a preceptor without receiving structured preceptor training or preparing themselves for the role, which also resulted in excessive work and stress [13,14]. A study conducted in Taiwan and the U.K. found that most nurses became preceptors without receiving sufficient training, and that the training they received was mostly theoretical rather than practical. Thus, it did not help them understand the demands of the preceptees and become good role models [15,16]. A study conducted in Korea revealed that the training and support systems for preceptors varied by the size of health care facilities. The facilities with fewer than 300 beds did not have standardized guidelines and had an inadequate number of personnel committed to nursing education, and such facilities often did not have a proper preceptorship support system [17]. However, detailed studies have not been conducted on the type or content of preceptor training programs. 

Previous studies of preceptor nurses focused mainly on the measurement or development of preceptors’ abilities by dealing with topics such as developing core competences or behavioral indexes, and did not pay enough attention to reviewing the status of preceptor training at each health care facility and examining the impact of such trainings. Thus, studies to prepare research materials regarding the need to develop preceptor training programs are needed. This study conducted a survey of nurses who had been preceptors, working in a variety of health care facilities, and investigated their experiences as a preceptor, whether they were provided with preceptor training programs, and the type of training with the aim of explaining differences among different groups of nurses. In addition, this study also examined the difference in the quality of preceptor training programs by analyzing the training satisfaction of preceptor nurses depending on their general characteristics.

Understanding the effect of preceptor training programs requires an understanding of preceptor competences. The concept that includes multiple competences is “clinical teaching behaviors”. Clinical teaching behaviors are the verbal and non-verbal interactions and actions of clinical instructors and preceptors, which facilitate learning achievements in clinical settings [18]. Clinical teaching behavior includes the commitment to teaching, building a learning atmosphere, using appropriate teaching strategies, guiding inter-professional communication, providing feedback and evaluation, and showing concern and support [19]. Effective clinical teaching behavior involves a series of purposeful actions to promote the transformation of theory to clinical practice and quick and effective work in graduate students [20]. Studies have found that preceptors with efficient teaching behaviors increase the adjustment, confidence, job satisfaction, and retention rate of novice nurses and help them to improve their unit-specific competence and decrease transition shock [20,21,22]. 

The clinical teaching behaviors of a preceptor can vary depending on the number of times a nurse takes on the preceptor role and the type and duration of the preceptorship. We can expect preceptor training to improve preceptors’ clinical teaching behaviors by clarifying their role and providing preceptors with knowledge and skills. Therefore, this study aimed to investigate the impact that the provision of such training has on the clinical teaching behaviors of preceptors. The purposes of this study were: (1) to explain the difference in preceptors’ experiences in training new nurses and the differences in the preceptor training courses depending on general characteristics; (2) to analyze the level of their clinical teaching behaviors and preceptors’ perceptions of their experience of preceptors and examine the difference in the clinical teaching behavior level according to characteristics of the preceptors; and (3) to divide nurses according to those who received preceptor training and those who did not, and analyze the factors that influenced their clinical teaching behaviors.

## 2. Materials and Methods

### 2.1. Design

This study was a descriptive cross-sectional study on nurses with experience as preceptors. We reviewed the preceptorship training experience of the nurses by the type of health care facilities and departments in which they worked, and identified whether the nurses received preceptor training or not. Then, we conducted a survey to investigate the impact of these factors on their teaching behaviors.

### 2.2. Participants

The participants included those who were working as nurses and had experience in training novice nurses as preceptors once or more at health care facilities at the hospital level or higher. The required number of participants was 166 when the effect size was set at 0.15, the significance level at 0.05, the power at 0.95, and the number of independent variables at 9 using G*Power 3.1.9.2. We selected a total of 180 participants considering the survey collection and response rates. All 180 respondents provided valid responses to all questions; therefore, all responses were included in our analysis. 

### 2.3. Instruments 

#### 2.3.1. General Characteristics of Participants 

We asked the nurses who had experience as preceptors about the type of healthcare facility in which they worked, the department in which they worked, and the length of their work experience as nurses. Then, we categorized them as nurses at tertiary hospitals and others in terms of the type of health care facility and as nurses who worked in a general ward, an intensive care unit (ICU), and others based on the department in which they worked. When it came to work experience, we divided the nurses into those with under five years of experience and those with experience of five years or more. 

#### 2.3.2. Preceptors’ Experience of Training New Nurses, Preceptor Training Course, and Their Satisfaction Level of Training Courses

We classified the number of precepting experience as one, two to three, and more than four times. With reference to the type of preceptorship, we asked about the ratio of preceptor to preceptees, and categorized preceptorships with a 1:1 or 2:1 ratio as good preceptorship, and preceptorships where one preceptor trained more than one preceptee at the same time as being worse than a 1:1 preceptorship. We categorized the duration of the most recent preceptorship as less than four weeks and four weeks and longer. Regarding whether they received preceptor training, we used their responses to yes or no questions. Regarding the type of preceptor training, we categorized the training type into didactic, simulation, basic nursing skills practice, and role-playing, and asked the respondents which type of training they attended and used the result in the analysis. For training satisfaction, we asked a question that used a Likert scale ranging from 1–7, where 1 meant “Not satisfied at all”, and 7 meant “Very satisfied”.

#### 2.3.3. Positive Perceptions of the Role of a Preceptor

To use the perceptions of the role of preceptors as a control variable, we referred to the results of a study conducted in Korea that interviewed preceptors about their positive perceptions of their experience as preceptors [23]. Then, we asked questions about the respondents’ perceptions using a Likert scale ranging from 1–7, where 1 was “Do not agree at all” and 7 was “Strongly agree”. We asked six questions, including ones stating that “Being a preceptor is an opportunity to gain knowledge”, “Being a preceptor motivates self-growth”, and “Being a preceptor is an opportunity to be credited for my work by others”. The Cronbach’s alpha was 0.83. 

#### 2.3.4. Clinical Teaching Behaviors

We measured the clinical teaching behaviors of preceptors using 22 questions from the Korean version of the Clinical Teaching Behavior Inventory [24], which was prepared by translating the Clinical Teaching Behavior Inventory (CTBI-23) developed by Lee-Hsieh et al. [19] and testing its validity and reliability. We acquired permission to use this tool from the corresponding author. The questions used a Likert scale ranging from 1–5, where 1 was “Strongly disagree” and 5 was “Strongly agree”. The Cronbach’s alpha was 0.93 in the previous study and 0.92 in this study. 

### 2.4. Data Collection 

We posted a survey recruitment letter on a social media platform used by many nurses. We also posted a URL where respondents could answer the questionnaire. Nurses who wanted to participate in this study could read the study description and fill out the consent form. We instructed the nurses to review their eligibility and check a box that said “I agree to participate in the research” online if they wanted to take part in the study. After receiving their consent, we provided them with the online questionnaire. The recruitment letter was posted on April 8, and the survey was conducted until May 27, 2020. 

### 2.5. Data Analysis 

The data were analyzed using SPSS version 23.0 (IBM Corp., Armonk, NY, USA). We used descriptive statistics such as frequency and percentage and conducted a Chi-squared analysis regarding the general characteristics of the participants and the difference in their experiences as preceptors. For the preceptor training experience and program satisfaction, the perceptions of the experience as preceptors, and the difference in clinical teaching behaviors, we used descriptive statistics such as the means and standard deviations and conducted a *t*-test or one-way ANOVA, then used the Scheffé’s method. We used Pearson’s correlation coefficient to analyze the correlation between perceptions of the experience as preceptors and clinical teaching behaviors and conducted a multiple regression analysis to investigate factors influencing the clinical teaching behaviors of preceptors.

### 2.6. Ethical Considerations

We collected the data for this study after acquiring the permission of the Institutional Review Board to which the researchers belong (SMU-2020-03-005), after informing them of the purpose of the research and appropriateness of the research method. 

## 3. Results

### 3.1. General Characteristics of the Participants and Their Experiences as Preceptors

The general characteristics of the participants and their experiences as preceptors are shown in Table 1. The number of respondents who worked at tertiary hospitals was 58.5% of the total 180 respondents. About 89.5% of the tertiary hospital nurses engaged in preceptorships where the ratio of preceptor to preceptee was 1:1 or 2:1, which was a ratio with even less pressure on preceptors compared to 1:1, while 64.0% of the nurses at other types of hospitals engaged in preceptorship with the same preceptor to preceptee ratios. The difference between the two groups was significant (χ^2^ = 17.11, *p* < 0.001). In addition, 63.8% of the tertiary hospital nurses engaged in preceptorship training for more than four weeks, while only 45.3% of the nurses at other type of hospitals engaged in training for more than four weeks (χ^2^ = 6.06, *p* = 0.014). When asked if they received preceptor training, 73.3% of the nurses at tertiary hospitals said yes, while only 42.7% of the nurses at other types of hospitals said yes, and the difference between the two groups was significant as well (χ^2^ = 17.23, *p* < 0.001).

When we divided the respondents by their department, it was found that 64.4% of them worked in a general ward, 22.8% in an ICU, and 12.8% in other departments. The percentage of ICU nurses who participated in preceptorship training with the same preceptor to preceptee ratio was 92.7%, which was the highest among the three groups, and 78.3% for nurses in other departments (χ^2^ = 6.26, *p =* 0.044). The percentage of nurses with the same duration of preceptorship training was 75.6% for the ICU nurses and 73.9% for the nurses in other departments. The difference among the three groups was significant (χ^2^ = 14.41, *p =* 0.001). When it came to the length of work experience as a nurse, 51.1% of the nurses had worked for five years or more and 48.9% had worked for less than five years. In total, 18.5% of the nurses with five or more years of experience had taken on the preceptor role once; 50.0%, two to three times; and 31.5%, more than four times. The difference in the number of times that the nurses played the role of preceptor between nurses with five or more years of experience and those with less than five years of experience was significant (χ^2^ = 30.27, *p* < 0.001). Additionally, 48.9% of nurses with fewer than five years of work experience received preceptor training, while 71.7% of nurses with five or more years of work experience received training, which shows the difference between the two groups (χ^2^ = 9.85, *p* = 0.002).

### 3.2. Training Satisfaction by the Type of Program

The general characteristics of 109 nurses who received preceptor training and their training satisfaction by the type of program are shown in Table 2. In total, 70.6% of nurses who received preceptor training were tertiary hospital nurses, and their satisfaction level was 4.39. This was significantly higher than the satisfaction level of nurses at other types of hospitals, which was 3.88 (*p* = 0.040). In terms of the type of program, 94.5% of the nurses who experienced a preceptor training program received didactic-based training. The percentage of nurses who received simulation-based training was 22.0%; core basic nursing skill education, 24.8%; and role-playing format, 16.5%. The difference in the satisfaction level by type of training was significant only for the core basic nursing skills practice (*p* = 0.019).

### 3.3. Clinical Teaching Behaviors

We asked a total of 22 questions about clinical teaching behaviors, and the overall value was 89.30 (Table 3). Among the six domains of clinical teaching behaviors, the mean score of “commitment to teaching” was the highest at 4.15, and the mean score of “guiding inter-professional communication” was the lowest at 3.94. The mean score for positive perceptions of the role of a preceptor was 4.87. The mean score of the question stating that “My perception that a preceptor should be a role model increased” was 5.47, which was the highest score, and the mean score of the question stating that “I felt proud of the growth of my preceptee(s)” was 5.42, which was the second-highest score. The mean score of the question asking whether “there was sufficient reward” was the lowest at 3.17. 

### 3.4. The Differences in Clinical Teaching Behaviors Depending on the General Characteristics

The differences in clinical teaching behaviors depending on the general characteristics of the participants and their experiences as preceptors are shown in Table 4. The difference, depending on the number of times that a nurse took on the role of preceptor, was significant; the mean score of the nurses who took on the preceptor role once was 87.40 and those who took on the preceptor role four times or more was 93.50 (*p* = 0.015). When the nurses were categorized by the type of preceptor training they received, the mean clinical teaching behavior scores of nurses who received role-playing-based preceptor training was 94.28. This was statistically significantly higher than the mean score of the nurses who did not receive the same type of training, which was 88.75 (*p* = 0.026). The clinical teaching behaviors of a preceptor had a positive correlation with positive perceptions of their experience as a preceptor. 

### 3.5. Factors Influencing Clinical Teaching Behavior According to Whether Preceptors Were Educated for a Preceptor Role

We divided the nurses into those who received preceptor training and those who did not and conducted a multiple regression analysis to examine the factors that have an influence on clinical teaching behaviors (Table 5). For those who did not receive preceptor training, the number of times they took on the role of preceptor, the type of preceptorship program, and the duration of such programs, did not have a significant influence on their clinical teaching behaviors. Those with more positive perceptions of the role of a preceptor had a higher clinical teaching behavior score (*p* < 0.001). Meanwhile, for the nurses who received preceptor training, the clinical teaching behavior score was higher for those who played a preceptor role more than four times than those who played the role just once (*p* < 0.001), and the level of clinical teaching behaviors was higher for those who received preceptor training in a role-playing format (*p* = 0.023). In addition, more positive perceptions of the role of a preceptor were associated with a higher clinical teaching behavior score (*p* < 0.001).

## 4. Discussion

This study was conducted to review the characteristics of nurses who had been preceptors and assigned to teach new graduate nurses; to examine the differences in their participation in preceptor training programs and in their experiences as preceptors; and to investigate the influence of their characteristics and experiences on their clinical teaching behaviors. Firstly, the ratio of preceptor to preceptee was 1:1 or better for the tertiary hospital nurses who played the role of preceptor, while the ratio was worse for nurses working at other types of hospitals (general hospital or hospital). A higher percentage of tertiary hospital nurses were provided with preceptor training compared to their counterparts. This finding is similar to the results of previous studies, which found that the level of support and training for preceptors varied depending on the number of beds of healthcare facilities [17]. The general hospitals and hospitals should be provided more support by sharing preceptor training methods and guides for clinical practice to improve the competences of preceptors and prevent their exhaustion while playing the role of preceptor. Moreover, only 24.8% of the participants had received core basic nursing skills practice education, which means that other health care facilities need to provide more of such education. Previous studies revealed that providing nurses with training that prepares them to be preceptors had a positive impact on them in terms of improving their knowledge, attitude, and skills as preceptors; therefore, more training programs for preceptors should be provided [7]. 

The domain with the highest mean score for clinical teaching behaviors was “commitment to teaching”, and the domain with the lowest mean score was “guiding inter-professional communication”. This result was different from that of a study conducted at a hospital in Hong Kong, where the domain with the highest mean score was “using an appropriate teaching strategy” and the domain with the lowest mean score was “providing feedback and evaluation” [25]. The participants of this study felt incompetent when they taught newly graduated nurses the skills for communicating with patients or other health care professionals. A previous study found that a preceptorship can have a critical impact on preceptees’ communication with patients or other health care professionals [26]; therefore, training in such skills should be included in preceptor training. Moreover, the clinical teaching behavior score was higher for nurses who took on the preceptor role more than four times compared to those who took on the role just once, which means that first-time preceptors can have more difficulties. Therefore, enough time and training should be provided to prepare them to be preceptors. 

Regarding perceptions of the experience of being a preceptor, the score was as low as 3.17 for the question asking whether the respondents received sufficient reward for playing the role of a preceptor. In a previous study about the experience of preceptor nurses in training novice nurses, the preceptors complained about excessive work and the exhaustion it caused [15,27]. The Ministry of Health and Welfare of Korea has been carrying out a personnel expenses subsidization project to encourage healthcare facilities to hire more nurses who would commit to training novice nurses [28]. Practical solutions, such as this project, and the participation of health care facilities in such solutions are required. 

Regarding the factors that influenced clinical teaching behaviors, we categorized the participants depending on whether they received preceptor training or not and conducted a multiple regression analysis. We found no difference in clinical teaching behaviors by nurses who did not receive preceptor training, regardless of the number of times they played the role of preceptor. On the contrary, for those who received preceptor training, clinical teaching behavior scores were higher for nurses who had played the preceptor role more than four times than for nurses who took on the role only once. This indicates that preceptors’ clinical teaching behaviors cannot be improved even if they take on the role of preceptor for a greater number of times without receiving training that can improve their competences as a preceptor. Even for nurses who have already been preceptors, appropriate preceptor training should be provided to improve their competences. In addition, we found that training through role-playing had a positive influence on clinical teaching behaviors, and that practicing the role of preceptor was an effective way to learn, which should be taken into consideration when developing preceptor training programs. The domain with the lowest score for preceptors’ clinical teaching behavior was “guiding inter-professional communication”. Such a skill could be taught effectively through role-playing; therefore, training programs in the role-playing format need to be developed for this domain. This study included the type of training in the analysis but did not examine the specific content of the training. Future research should not only include the type of training but also its content in the analysis and examine the impact of the training content in detail. 

We also found that nurses with more positive perceptions of the role of preceptor had a higher score for clinical teaching behavior, which indicates that providing preceptors with continuous encouragement, acknowledgment, and proper rewards for their work and efforts could further improve their competences. A higher level of clinical teaching behavior can facilitate the socialization and job satisfaction of preceptees as well, which would result in a more stable workforce and the improvement of the competences of all nurses. 

This study has a few limitations. The causal relationships might be unclear because we used a cross-sectional research method. Regarding the training that preceptors received, we collected information on the type of training but not the training content. Future research should analyze the training content along with the training type and examine the effect of the content. 

## 5. Conclusions

We examined the differences in nurses’ experiences as preceptors and their participation in preceptor training programs depending on the type of health care facility and the department in which they worked, and investigated the influence of such characteristics and experiences on their clinical teaching behaviors. We found that general hospitals or hospitals still lack structured training programs for preceptors compared to tertiary hospitals, which calls for more policy support. In addition, health care facilities should actively make use of policy supports such as the government’s project to subsidize medical institutions for hiring nurses who can commit to training to reduce the excessive workload for preceptors. We also found that more experience as a preceptor can improve preceptors’ clinical teaching behaviors only if preceptor training is offered. Future research should put more effort into analyzing the content of preceptor training programs and developing preceptor training programs that can improve the clinical teaching behaviors of preceptors. 

## Figures and Tables

**Table 1 ijerph-18-00975-t001:** Descriptive statistics of participants’ characteristics and experiences as preceptors (*n* = 180).

Characteristics	Categories	*n* (%)	No. of Precepting Experiences				Ratio of Preceptor to Preceptee			Duration of Preceptorship (Weeks)			Training for Preceptorship	
			1	2–3	≥4	χ^2^ (*p*)	1:1 or Better	Worse Than 1:1	χ^2^ (*p*)	<4	≥4	χ^2^ (*p*)	Yes	χ^2^ (*p*)
Type of institution	Tertiary hospitals	105 (58.3)	35 (33.3)	49 (46.7)	21 (20.0)	0.26 (0.880)	94 (89.5)	11 (10.5)	17.11 (<0.001)	38 (36.2)	67 (63.8)	6.06 (0.014)	77 (73.3)	17.23 (<0.001)
	Others	75 (41.7)	27 (36.0)	35 (46.7)	13 (17.3)		48 (64.0)	27 (36.0)		41 (54.7)	34 (45.3)		32 (42.7)	
Department	General ward	116 (64.4)	38 (32.8)	51 (44.0)	27 (23.3)	4.68 (0.322)	86 (74.1)	30 (25.9)	6.26 (0.044)	63 (54.3)	53 (45.7)	14.41 (0.001)	72 (62.1)	0.34 (0.845)
	Intensive care unit	41 (22.8)	14 (34.2)	22 (53.7)	5 (12.2)		38 (92.7)	3 (7.3)		10 (24.4)	31 (75.6)		24 (58.5)	
	Others	23 (12.8)	10 (43.5)	11 (47.8)	2 (8.7)		18 (78.3)	5 (21.7)		6 (26.1)	17 (73.9)		13 (56.5)	
Work experience as a nurse (year)	<5	88 (48.9)	45 (51.1)	38 (43.2)	5 (5.7)	30.27 (<0.001)	65 (73.9)	23 (26.1)	2.61 (0.106)	41 (46.6)	47 (53.4)	0.5 (0.475)	43 (48.9)	9.85 (0.002)
	≥5	92 (51.1)	17 (18.5)	46 (50.0)	29 (31.5)		77 (83.7)	15 (16.3)		38 (41.3)	54 (58.7)		66 (71.7)	

**Table 2 ijerph-18-00975-t002:** Experiences of preceptorship training and satisfaction with training courses (*n* = 109).

Category	Variable		*n* (%)	M ± SD	*t* or F (*p*)
Type of institution	Tertiary hospitals		77 (70.6)	4.39 ± 1.26	2.08 (0.040)
	Others		32 (29.4)	3.88 ± 0.94	
Department	General ward		72 (66.1)	4.25 ± 1.15	0.07 (0.935)
	Intensive care unit		24 (22.0)	4.17 ± 1.09	
	Others		13 (11.9)	4.31 ± 1.65	
Work experience as a nurse (year)	<5		43 (39.5)	4.12 ± 1.18	0.86 (0.391)
	≥5		66 (60.5)	4.32 ± 1.20	
Education type	Didactic	Yes	103 (94.5)	4.26 ± 1.20	0.54 (0.461) *
		No	6 (5.5)	3.83 ± 1.17	
	Simulation	Yes	24 (22.0)	4.50 ± 1.18	1.22 (0.226)
		No	85 (78.0)	4.16 ± 1.19	
	Nursing skill practice	Yes	27 (24.8)	4.70 ± 1.14	2.39 (0.019)
		No	82 (75.2)	4.09 ± 1.18	
	Role-playing	Yes	18 (16.5)	4.28 ± 1.45	0.15 (0.880)
		No	91 (83.5)	4.23 ± 1.15	

* Mann–Whitney test.

**Table 3 ijerph-18-00975-t003:** Clinical teaching behavior and positive perceptions of precepting experiences (*n* = 180).

Categories	Items	M ± SD
Clinical teaching behavior	Overall (range: 22–110)	89.30 ± 10.00
	Committing to teaching	4.15 ± 0.51
	Building a learning atmosphere	4.11 ± 0.52
	Using appropriate teaching strategies	4.07 ± 0.55
	Guiding inter-professional communication	3.94 ± 0.66
	Providing feedback and evaluation	4.00 ± 0.64
	Showing concern and support	4.05 ± 0.61
Positive perceptions of precepting experiences	Total	4.87 ± 0.98
	Strengthened nursing practice knowledge	5.08 ± 1.30
	Self-growth	5.28 ± 1.25
	Opportunity to be recognized	4.76 ± 1.27
	Pride in the growth of preceptee	5.42 ± 1.30
	Increased awareness of being a role model	5.47 ± 1.30
	Sufficient reward	3.17 ± 1.49

**Table 4 ijerph-18-00975-t004:** Clinical teaching behavior according to general characteristics and correlations with perceptions of precepting experiences (*n* = 180).

Variables	Categories		*n* (%)	Clinical Teaching Behavior	
				M ± SD or r	*t* or F (*p*)
No. of precepting experiences	1 ^(a)^		62 (34.4)	87.40 ± 9.55	4.29 (0.015) a < c
	2–3 ^(b)^		84 (46.7)	89.00 ± 9.58	
	≥4 ^(c)^		34 (18.9)	93.50 ± 10.94	
Ratio of preceptor to preceptee	1:1 or better		142 (78.9)	89.26 ± 9.63	−0.10 (0.919)
	Worse than 1:1		38 (21.1)	89.45 ± 11.47	
Duration of preceptorship (weeks)	<4		79 (43.9)	89.97 ± 10.65	0.80 (0.426)
	≥4		101 (56.1)	88.77 ± 9.51	
Training for Preceptorship	Yes		109 (60.6)	89.51 ± 9.94	0.35 (0.724)
	No		71 (39.4)	88.97 ± 10.20	
Education type	Didactic	Yes	103 (57.2)	90.05 ± 9.28	1.16 (0.247)
		No	77 (42.8)	88.30 ± 10.91	
	Simulation	Yes	24 (13.3)	86.83 ± 13.26	−1.30 (0.196)
		No	156 (86.7)	89.70 ± 9.41	
	Nursing skills practice	Yes	27 (15.0)	91.07 ± 10.97	1.00 (0.319)
		No	153 (85.0)	88.99 ± 9.85	
	Role-playing	Yes	18 (10.0)	94.28 ± 10.71	2.25 (0.026)
		No	162 (90.0)	88.75 ± 9.81	
Positive perception of precepting experiences				0.332	<0.001

**Table 5 ijerph-18-00975-t005:** Factors influencing clinical teaching behavior based on whether the nurse was educated for a preceptor role (*n* = 180).

	Non-Educated Preceptor	Educated Preceptor
Number of precepting experiences (vs. 1)		
2~3	−0.09 (0.436)	0.11 (0.239)
≥4	0.04 (0.821)	0.31 (<0.001)
Ratio of preceptor to preceptee (vs. preceptor:preceptee = 1:1 or better)	0.25 (0.058)	−0.14 (0.212)
Duration of preceptorship (vs. <4 weeks)	−0.17 (0.137)	−0.03 (0.714)
Education type: didactic (vs. yes)		0.33 (0.058)
Education type: simulation (vs. yes)		−0.19 (0.070)
Education type: nursing skills practice (vs. yes)		0.12 (0.242)
Education type: role-playing (vs. yes)		0.25 (0.023)
Positive perception of precepting experiences	0.48 (<0.001)	0.14 (<0.001)
	Adj R^2^ = 0.202, F = 4.54	Adj R^2^ = 0.254, F = 5.08

## Data Availability

The data presented in this study are available on request from the corresponding author.

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
