# Peer review of "Effect of Nurses’ Preceptorship Experience in Educating New Graduate Nurses and Preceptor Training Courses on Clinical Teaching Behavior"

_ijerph, 2021, doi:10.3390/ijerph18030975_

Round 1

Reviewer 1 Report

Thank you very much for the opportunity to review this manuscript. I do have some suggestions for needed edits that would assist in adding clarity to the paper.

Some English language edits are needed throughout including rewording, correcting run-on sentences, checking spelling, and adding missing words/phrases:

Line 12, 42, 83, 92, 100, 119-120 (duplicate sentence), 125-127, 357-359, spelling of word competencies throughout, Table 1 (type of preceptorship label needs to be edited).

Please be consistent when describing the types of facilities throughout manuscript. As an example, some areas describe facilities as "hospitals" or "lower-level facilities" (unsure of what these are); other areas describe as "larger" and "smaller-size" facilities. What is "hospital level or higher" mean?

Early on, clarify the type of preceptorship this is referencing as some preceptorships take place during nursing education versus as a new graduate.

Provide further detail regarding the reference that delineates the clinical teaching behaviors included in the survey as this is not clear. 

Line 18 - is this the overall average score? Need to clarify. 

Reviewer 2 Report

Thanks for this important manuscript.

Nurses are the most stable component in the emergency system.  While physicians come and go, it is the experienced nurses who have been there for years to “remind” us physicians what should be done when crisis’s strikes.

I enjoyed reading this well-written paper with no major errors. The only drawback is the lack of international literature. In such a study, 23 references are definitely insufficient.

Following you will find best new papers contected with nursing percepction and learning that I think should be included in your study. Beside these please also improve references by adding more studies

  • Lee JH, Noh HR, Park JY. Predictive Factors of Turnover Intention in New Nurses. Journal of muscle and joint health. 2020;27(1):50-60.
  • Goniewicz K, Goniewicz M, Burkle FM, Khorram-Manesh A. Cohort research analysis of disaster experience, preparedness, and competency-based training among nurses. PloS one. 2021 Jan 8;16(1):e0244488.
  • Tayebi Z, Lotfi R, Tayebi Arasteh M, Amiri S, Shiri M. The Investigation of Effect of Preceptorship Program on Promoting Practical Skills of Nursing Students in Alborz University of Medical Sciences: An Action Rresearch Study. Alborz University Medical Journal. 2020 Jun 10;9(3):269-76.
  • Walker SH, Norris K. What is the evidence that can inform the implementation of a preceptorship scheme for general practice nurses, and what is the evidence for the benefits of such a scheme?: A literature review and synthesis. Nurse Education Today. 2020 Mar 1;86:104327.

additional comments:

This is actually one of the better papers I have read. And I used to review a lot for many journals just like being Academic Editor for Plos and BMC. The results match the summary of results presented in the abstract and align with the aim. Data is presented in an appropriate way, table and figures are relevant and clearly presented. The appropriate units, rounding, and number of decimals were considered; the table title, columns, and rows are correctly and clearly labeled. This paper offers novel new insights and in this respect, the manuscript reviewed can be published. The one more concern I could raise is similar to other reviewers. The authors could describe more specifically the facilities that were used in the manuscript providing more details in description

Author Response

Thank you for you comment. Please see the attachment.
